

# Impact of basement thrust fault on low-angle normal fault and rift basin evolution: a case study in the Enping sag, Pearl River Mouth Basin

Chao Deng[1], Rixiang Zhu[2, 1], Jianhui Han[3], Yu Shu[4], Yuxiang Wu[4], Kefeng Hou[5], Wei Long[3]

[1] State Key Laboratory of Continental Dynamics, Department of Geology, Northwest University, Xi'an 710069, China.
[2] Beitucheng West Road, Institute of Geology and Geophysics, Chinese Academy of Sciences, Chaoyang District, Beijing, 100029, China.
[3] College of Energy, Chengdu University of Technology, Chengdu, Sichuan 610059, China.
[4] Shenzhen Branch Company of CNOOC, Shenzhen, China.
[5] Changqing Oil Field, PetroChina Company Limited, CNPC, Xi'an, 710021, China.

*Correspondence to*: Chao Deng (dengchao@nwu.edu.cn), Rixiang Zhu (rxzhu@mail.igcas.ac.cn)

**Abstract.** Reactivation of pre-existing structures and their influence on subsequent rift evolution have been extensively analysed in previous researches on rifts that experienced multiple phases of rifting, where pre-existing structures were deemed to affect nucleation, density, strike direction and displacement of newly-formed normal faults during later rifting

stage. However, previous studies paid less attention to the extensional structures superimposing on an earlier compressional background, leading to a lack of understanding, e.g., reactivation and growth pattern of pre-existing thrust fault as low-angle normal fault, and impact of pre-existing thrust fault on newly-formed high-angle faults and subsequent rift structure. This study investigating the spatial relationship between intrabasement thrust and rift-related faults in the Enping sag, northern South China Sea, indicates that the rift system is built on the previously deformed basement with pervasive thrusting

structures, and that the low-angle major fault of the study area results from reactivation of intrabasement thrust fault. In addition, reactivated basement thrust fault can affect the nucleation and dip of nearby new faults, cause new faults to nucleate at or merge into it in vertical, and finally generate typical fault interactions, i.e., merging, abutting and crosscutting faults. This work not only provides insights into understanding growth pattern of rift-related faults interacting with reactivated low-angle faults, but also has broader implications on how basement thrust faults influence rift structure, normal

fault evolution and syn-rift stratigraphy.

## 1 Introduction

Reactivation of pre-existing basement faults or shear zones and their influence on the growth of newly-formed normal faults have been recognized in many multiphase rift basins, such as the NW Shelf of Australia (e.g., Frankowicz & Mcclay, 2010), Gulf of Thailand (e.g., Morley et al., 2004, 2007), Gulf of Aden (e.g., Lepvrier et al., 2002; Bellahsen et al., 2006), Northern

North Sea (e.g., Badley et al., 1988; Færseth, 1996; Færseth et al., 1997; Odinsen et al., 2000; Whipp et al., 2014; Duffy et





al., 2015; Deng et al., 2017a; Fazlikhani et al., 2017), and East African Rift (Le Turdu et al., 1999; Lezzar et al., 2002; Corti, 2009; Muirhead & Kattenhorn, 2017). Previous researches suggested that pre-existing basement faults or shear zones originated from an earlier tectonic event were prone to reactivate, interact and link with newly-formed normal faults during a later rifting stage, and finally resulted in some special structural styles, e.g., non-collinear fault arrays, various styles of fault

interactions, and gradual change in fault geometries with increased depth. In addition, our understanding of three-dimensional fault geometry and evolution affected by pre-existing basement weaknesses or faults has been greatly improved by a number of studies using physical analogue and numerical models (e.g., McClay and White, 1995; Keep and McClay, 1997; Henza et al., 2010, 2011; Deng et al., 2017b, 2018). Based on those researches, people notice that a pre-existing basement fault can reactivate in variable modes as a response of a changing strain magnitude and/or extension direction, and

play an important role on the geometry of neighbouring new faults, such as fault density, orientation and displacement (e.g., Morley et al., 2004, 2007; Henza et al., 2010, 2011; Deng et al., 2017b, 2018). However, previous studies stressed more attention to the rift basins that evolved through multiple phases of rifting, yet the characteristics of extensional structures developing after a period of compressional event, such as reactivation mode of pre-existing thrust faults, growth pattern of low-angle normal faults and their interactions with high-angle normal faults, still remain short of investigation.

The South China Sea, located at the junction area of the South China Block, Indo-China Block and Pacific Plate (Fig. 1), is known as a Late Cretaceous to Early Cenozoic rift system that was built on a previously deformed basement containing pre-existing thrust and strike-slip faults, due to negative inversion from compressional to extensional tectonic setting during the Late Mesozoic (e.g., Holloway, 1982; Taylor and Hayes, 1983; Li and Li, 2007; Li et al., 2012; Shi and Li, 2012). The influence of pre-existing basement structures on the rift basin evolution of the northern South China Sea has already been

reported by previous researchers. For example, Hu et al. (2013) argued that the tectonic evolution of the Qiongdongnan Basin was controlled by faults reactivated from NE-SW- and E-W-trending pre-existing fabrics. Also, Ye et al. (2020) suggested that the overall rift architecture in the proximal domain of the northern South China Sea margin is mainly controlled by reactivation of two pre-existing fault systems within the basement, such as the WNW- to EW- and the ENE-striking pre-existing faults. However, there are still some key issues concerning about the evolution of the fault network and

rift basins. For instance, how does pre-existing thrust fault reactivate and grow as a low-angle normal fault during the subsequent rifting? How does pre-existing thrust fault control the development of newly-formed normal faults and rift basins? Answering those questions will help to improve our understanding of the structural style and rift development over pre-existing basement thrusting structures in general.

In this contribution, the Enping sag in the Pearl River Mouth Basin is selected as the starting point for elucidating the

influence of pre-existing basement thrusting structures on normal fault and rift basin development (Fig. 1c). Firstly, we integrate the 3D seismic reflection and borehole data in the study area to interpret the geometry of basement thrust faults and their spatial relationship with rift-related normal faults. On this foundation, the kinematics of pre-existing basement structures and newly-formed normal faults, as well as their interactions are analysed according to throw variation information. The objective of this step is to figure out the growth and interaction styles between pre-existing basement thrust





faults and newly-formed normal faults, along with fault evolution of the study area. At the second step, we try to explore the reactivation mode of basement thrust faults and the factors controlling their reactivation and rift development in the northern South China Sea. Finally, we discuss about the impacts of basement thrust faults on the newly-formed fault growth and rift development during a subsequent rifting stage in general. Through detailed study of the basement and rift-related structures, this work will not only provide insights into understanding the fault growth pattern with interactions between high-angle rift-related faults and low-angle basement thrust faults, but also have broader implications on how basement thrust faults influence rift basin development.

## 2 Geological setting

The South China Sea is a tectonically complex area that has been affected by both collision of the Indo-China Block and subduction of the paleo-Pacific Plate toward the SW and SE boundaries of the South China Block, respectively (e.g., Charvet et al., 1994; Li, 2000; Zhou et al., 2006; J. Li et al., 2014; Z. Li et al., 2014). The pre-Cenozoic evolution of the South China Sea area was mainly controlled by two periods of regional tectonic movement: the Indosinian movement (251–205 Ma) and the Yanshanian movement (180–67 Ma) (Zhou et al., 2006). The Indosinian movement was dominated by the orogeny and suture between the South China Block and the Indochina Block, causing the formation of NW-SE-trending thrust and strike-slip faults at the western margin of the South China Block (Metcalfe, 2006). The Yanshanian movement was driven by NW subduction of the paleo-Pacific Plate beneath the South China Block, and was structurally characterized by E-W- to NE-SW-trending thrust and strike-slip faults (Faure et al., 1996; Shu et al., 2006; Zhou et al., 2006). At the end of the Yanshanian movement, the South China Block underwent a tectonic transition from compressional to extensional setting, which is supposed to result in the onset of continental rifting from c. 60Ma or even earlier (Holloway, 1982; Taylor and Hayes, 1983; Li and Li, 2007; Li et al., 2012; Shi and Li, 2012). During the Cenozoic, the South China Sea area experienced a long episode of intense continental rifting, breakup and seafloor spreading, producing numerous extensional basins that were infilled with a large volume of red fluvial and lacustrine sediments known as Red Beds (e.g., Li, 1999; Shu et al., 2009; J. Li et al., 2014).

Previous studies indicated that the Cenozoic rift evolution of the South China Sea was featured by episodic rifting, with a sequential clockwise rotation of the extension direction (e.g., Ru and Pigott, 1986; Pigott & Ru, 1994; Zhou et al., 1995). Three rifting episodes have been identified: i) the Late Cretaceous to Early Eocene rifting (~87-50 Ma), ii) the Early Eocene rifting (49-39 Ma), and iii) the Late Eocene to Early Oligocene rifting (39-32 Ma). The Late Cretaceous to Early Eocene rifting episode initiated some small half-grabens that were bounded by NNE- or NE-trending normal faults in response to a WNW-ESE extension and infilled with the Shenhu Formation comprising continental red beds with volcanic and metamorphic clasts. During the Early Eocene rifting episode, there had a clockwise rotation of the extension direction to NNW-SSE, leading to the development of NE- to ENE-trending normal faults and the deposition of the Wenchang Formation. Furthermore, the Late Eocene to Early Oligocene rifting episode had a further clockwise rotation of the extension



direction to nearly N-S, which caused the development of E-W-trending normal faults and the deposition of the Enping Formation. Rift activity ceased when continental breakup occurred and seafloor spreading began at ca. 32 Ma, and thereafter, the South China Sea entered the post-rift stage that was dominated by regional thermal subsidence (e.g., Li, 1993; Sun et al.,
2014; Wu et al., 2014).

The Enping Sag is an overall NE-SW-trending negative zone in the Pearl River Mouth Basin at the northern margin of the South China Sea (Fig. 1). The NW and SE boundaries of the Enping Sag are defined by the Northern Uplift Zone and Panyu Lower Uplift, respectively; and in the NE and SW, it gradually transfers to the neighbouring sags through NW-trending structures. In a smaller scale, the Enping sag is further divided into three sub-sags: Ep12, Ep17, and Ep18 sub-sags (Fig. 1c).
The Ep12 and Ep17 sub-sags are bounded in the northwest by the NE-SW-striking major fault F1, whereas the Ep18 sub-sag is bounded in the north by the E-W-striking major fault F2 (e.g., Xu et al., 2014; Shi et al., 2015). The syn-rift stratigraphic units include the Wenchang and Enping Formations (Fig. 2), but the Shenhu Formation is absent in the Enping Sag, possibly because of diachronous rifting from the east to the west (Franke et al., 2014; Savva et al., 2014; Morley, 2016). The Wenchang Formation that directly overlies the top basement mainly consists of grey to black organic-rich shale interbedded
with sandstone being deposited in middle to deep lacustrine, corresponding to the Early Eocene rifting episode (Episode 1, Fig. 2). The overlying Enping Formation is composed of shale, sandstone, and thin coal beds from fluvial to shallow and middle lacustrine, representing the deposition of the Late Eocene to Early Oligocene rifting episode (Episode 2, Fig. 2; Liu et al., 2016). Post-rifting strata start from the Zhuhai Formation, which gradually transforms to marine facies as a result of continental breakup (e.g., Li, 1993; Sun et al., 2014; Wu et al., 2014).

**3 Data and methods**

Structural interpretation and mapping are performed using the 3D seismic cube and wellbore data covering the study area (Fig. 1). The 3D seismic cube has an area of ~2300 km$^2$ with a line spacing of 12.5 m, and images down to ca. 6 s TWT in depth. The seismic reflection is displayed in normal polarity (SEG Convention), whereby a downward increase in acoustic impedance is indicated by a peak (brown reflection) and a decrease in acoustic impedance is represented by a trough (black
reflection) (Figs. 2 and 3). In addition, two wells are tied to the seismic section by means of synthetic seismograms, which is sufficient for tracing and mapping the horizons in the study area (Fig. 3). Well-18 drills through the syn-rift strata and penetrates into the basement, while the other wells stop within the syn-rift strata.

Three key seismic horizons, i.e., $T_g$, $T_{80}$ and $T_{70}$ from bottom to top, are mapped for establishing the overall structural framework (Fig. 2). Among them, $T_g$ horizon is a very strong and continuous reflector that separates the overlying layered
reflections from the underlying chaotic reflections, indicating that it is a regional angular unconformity representing the rift onset; thus, we define it as the top of the crystalline basement (Fig. 2; Ru and Pigott, 1986; Li et al., 1999; Yan et al., 2014). $T_{70}$ horizon is a strong reflector that is located close to the ceiling of the wedge-shaped growth stratigraphy, i.e., the $T_g$ to $T_{70}$ interval, representing the breakup unconformity formed in response to the continental breakup and incipient seafloor



spreading of the South China Sea; thus, we define it as the top of the syn-rift stratigraphic units (Fig. 2). $T_{80}$ horizon is
located at the interface between the Wenchang and Enping Formations according to the seismic-well tie, so is defined as a
local angular unconformity within the syn-rift stratigraphy that separates the Wenchang from the Enping Formations (Figs. 2
and 3).

The key structures developing in the study area are classified into two groups: the intrabasement structure and the rift-related
cover structure that are separated from each other by the top basement or $T_g$ horizon (Fig. 4). The intrabasement structure is
identified according to the strong seismic reflection packets that are constrained within the crystalline basement (Fig. 4b).
Because of limited lateral continuity of seismic reflectors within the basement, the intrabasement structure is mainly
portrayed in cross-sectional view, with the goal of displaying their geometric style and spatial relationship with the cover
structures. The rift-related cover structure, however, is primarily interpreted according to the offset of the continuous
reflectors in the stratigraphic package, and is described in regard to fault length, strike, dip and throw (Fig. 4c).
On the basis of the structural interpretation and mapping, we employ two commonly-used methods to analyse the fault
kinematics: i) throw-length (T-x) plots that show fault throw (T) against the distance (x) along fault strike are used to study
the propagation and linkage history of fault segments (e.g., Peacock and Sanderson, 1991; Cartwright et al.,1995; Dawers
and Anders,1995; Gupta and Scholz, 2000; Baudon and Cartwright, 2008); ii) throw-depth (T-z) profiles that display fault
throw (T) against depth (z) can be employed to interpret fault propagation history in vertical (e.g., Cartwright et al., 1998;
Baudon and Cartwright, 2008).

## 4 Structural style and fault geometry

### 4.1 Intrabasement structure

We discover two types of strong seismic reflections within the crystalline basement. The first type of intrabasement
reflections (Type I) is a relatively linear, thin (c. 10-100 ms TWT), high-amplitude reflection packet that is composed of a
trough-peak-trough wave assemblage in cross-section (Figs. 4 and 5). The second type of intrabasement reflections (Type II)
is characterized by a package of relatively winding, thick (c. 500-1000 ms TWT), medium- to high-amplitude reflections
(Fig. 5c).

Type I intrabasement reflections, e.g., BF1 in the hanging wall of the rift-related fault F1, BF2 and BF3 in the footwall of
fault F3, BF4 and BF5 in the footwall of fault F4, and BF6 in the footwall of fault F6 at the eastern border of the study area,
etc., are dominantly dipping at <30º toward SE, with a lateral spacing of c. 500 m between each other (Figs. 5 and 6d). Also,
most of Type I intrabasement reflections tip-out at or below the top basement, except that the upper tip of intrabasement
reflection BF2 exactly extends above $T_g$ horizon and offsets a few reflection waves in the stratigraphic cover (Fig. 5b). It is
uncertain about where the lower tip of Type I intrabasement reflections terminates due to poor seismic imaging at the depth
larger than 4 s TWT, but some of them may extend downward and connect with shear zones in the lower crust. In addition,
we observe that intrabasement reflection BF3 merges into the plane of fault F3 with increased depth, and has approximately





the same dip angle and seismic reflection characteristic with the lower part of fault F3 in the basement rock (Fig. 5b). These observations confirm that Type I reflections represent real geological boundaries rather than acquisition- or processing-related geophysical artefacts.

In comparison, a set of high-amplitude, folded or winded, Type II intrabasment reflections develop to the SE of Type I
reflection BF4 at the depth of c. 3-4 s TWT, resembling a fault propagation folding related to thrust faulting structure (Fig. 5c). In addition, we observe that the bottom of Type II reflections is displaced by some Type I intrabasement reflections, having an up-dip offset of c. 50 ms TWT (Fig. 5c). Similar characteristics are also observed in the footwall of fault F6, where a group of Type II intrabasement reflections are folded and displaced by Type I intrabasement reflections (Fig. 6d). Although the top and bottom of Type II reflections can be sketched in cross-section, it is too challenging to trace them across
the study area because they have bad lateral continuity.

Based on the observations described above, we interpret Type I intrabasement reflections to be pre-existing basement faults, i.e., basement faults BF1-BF6. According to the deformation manner of Type II intrabasement reflections, they are suggested to be pre-existing basement thrust faults originated from the compressional tectonics prior to the Late Cretaceous to Early Cenozoic extensional event. There are three evidences supporting our interpretation: i) termination of the upper tip of Type I
intrabasement reflections against or beneath the top basement, such as BF1 and BF3, indicating that they are pre-existing faults forming before the initiation of the Late Cretaceous to Early Cenozoic rifting stage; ii) fault propagation folding structure that occurs close to the basement faults BF4 and BF6, implying that folding structure is the immediate result of reverse fault movement; and iii) reverse offset of Type II intrabasement reflections across Type I intrabasment reflections, which is a good representative of up-dip slip of reverse fault (Figs. 5 and 6). For other basement faults, it is difficult to
measure their displacement owing to the lack of traceable strong reflections across them; thus, it is less possible to determine their sense of shear. Based on the aforementioned evidences, we suggest that there exist a number of NE-SW- or E-W-striking thrust faults in the basement of the Enping sag, which are supposed to be associated with the compressional event prior to the Late Cretaceous to Early Cenozoic rifting. This suggestion is consistent with previous studies in relation to the development of basement thrust faults (e.g., Nanni et al., 2017; Ye et al., 2020).

**4.2 Rift-related cover structures**

The Late Cretaceous to Early Cenozoic rift structure is dominated by two major faults that are >15 km in length and >200 ms TWT of maximum throw, namely the NE-SW-striking major fault F1 and the approximately E-W-striking fault F2 lying in the hanging wall of fault F1 (Fig. 7). The western end of the major fault F2 terminates against the middle part of major fault F1, exhibiting a connected fault array at the top basement. The major faults bound three separated syn-rift half-grabens, i.e.,
the Ep12, Ep17 and Ep18 sub-sags, in the hanging wall of the eastern and western parts of fault F1, and fault F2, respectively. Apart from the major faults, there also develop a few numbers of minor faults that are <15 km in length or <200 ms TWT of maximum throw in the hanging wall of the major faults (Figs. 6 and 7). Following sections are detailed description of the geometry and structural relationship between the major and minor faults.



### 4.2.1 Major faults

The major fault F1 has a length of c. 60 km and a dominant NE-SW strike, but makes a bend as it turns to nearly N-S at its westernmost end (Fig. 7). In addition, the cross-sectional geometry of the major fault F1 and the corresponding hanging-wall stratigraphy significantly changes along strike. In the southwest, for example, fault F1 is at ~10° dips toward SE and shows a 'ramp-flat' shape in cross-section view (Fig. 6a). The upper tip of fault F1 terminates close at $T_{70}$ horizon, while the lower tip penetrates into the basement and extends southward beyond the study area, making a vertical offset of c. 1500 ms TWT at

$T_g$ horizon (Fig. 7). In the hanging wall of fault F1, i.e., the Ep17 sub-sag, the NW-expanding stratigraphic wedge between $T_g$ and $T_{70}$ horizons is c. 1700 ms TWT thick and c. 35 km wide in maximum. In the central part, however, the shape of fault F1 is approximately listric from top to bottom, with a gentle dip at c. 10° (Fig. 6b). In addition, the upper tip of fault F1 reaches above $T_{70}$ horizon, and its lower tip extends into the basement, creating a vertical offset of c. 1000 ms TWT at $T_g$ horizon (Fig. 7). The stratigraphic interval between $T_g$ and $T_{70}$ horizons, c. 1200 ms TWT of thickness in maximum, is

featured by a package of sub-tabular stratigraphy that gradually overlaps the slipping plane of fault F1. In the northeast part of fault F1 where the Ep12 sub-sag is located, the geometries of the fault plane and hanging-wall stratigraphy are in sharp contrast with the previous parts. Firstly, fault F1 shows a 'dogleg' shape in cross-section, with its upper portion at c. 15° dips and the lower portion at c. 10° dips (Fig. 6c). Secondly, the upper tip of fault F1 terminates at $T_{80}$ horizon, and its lower tip extends downward into the basement, making a vertical offset of c. 800 ms TWT at Tg horizon (Fig. 7). Thirdly, the $T_g$ to

$T_{80}$ stratigraphic interval in the hanging wall of fault F1 is also characterized by a wedge-shaped growth package, but it is only c. 800 ms TWT thick and c. 8 km wide in maximum. Fourthly, significant rollover or rotation of stratigraphic beds occurs in the hanging-wall block, accompanied by erosion of the $T_g$ to $T_{80}$ stratigraphy at the crest of the fault block. Finally, the overlying $T_{80}$ to $T_{70}$ stratigraphic package, only c. 300 ms TWT of thickness, is approximately tabular and unaffected by fault F1 (Fig. 6c). Going further east, there develops an intrabasement fault BF6 of which the position coincides with the

northeast extension line of the major fault F1. However, the top basement and the stratigraphic beds above intrabasement fault BF6 have no offset, suggesting that the major fault F1 does not propagate into this section (Fig. 6d). In the footwall of fault F1, there is no much difference in the stratigraphic pattern along strike, because the top basement is almost horizontal and overlain by a tabular stratigraphic package. As a whole, we observe a NE-trending decrease in the final offset of the major fault F1 (Fig. 7). Also, the thickness and width of the overlying syn-rift stratigraphy also decrease from SW to NE.

Such a smaller volume of the syn-rift stratigraphy in the Ep12 sub-sag compared to the Ep17 sub-sag is possibly related to the synchronous growth of fault F2 with fault F1 in the eastern part of the Enping sag.

The major fault F2 is c. 20 km long, nearly E-W-striking and S-dipping in map view (Fig. 7). Fault F2 is situated in the hanging wall of major fault F1, with its western tip terminating against the major fault F1 and its eastern tip extending away from the major fault F1. In cross-section, fault F2 has a steeper dip than fault F1, with a listric shape from top to bottom (Fig.

6c). In details, the upper part of fault F2 is at c. 45° dips and extends upward to the surface of the stratigraphic beds, whereas its lower part gradually becomes gentler and ends up joining into the more shallowly-dipping major fault F1 at depth. In





addition, the $T_g$ to $T_{70}$ stratigraphic interval in the hanging wall of fault F2 is characterized by a thick wedge-shaped growth package that has a maximum thickness of c. 1600 ms TWT. In contrast, the stratigraphic package above $T_{70}$ horizon has an approximately tabular shape (Fig. 6c). Similar to the northeast part of fault F1, prominent rollover and/or rotation of the

stratigraphic package is also observed in the hanging wall of fault F2.

### 4.2.2 Minor faults

Minor faults, c. 5 km long, E-W-striking and at ~60° dips in general, are mostly distributing in the hanging wall of the major faults F1 and F2 (Fig. 6). Different from the major faults, minor faults mainly occur in the stratigraphic cover. For instance, the upper tip of the majority of the minor faults extends above $T_{70}$ horizon, and the lower tip of them terminates above the

top basement, i.e., fault F5 (Fig. 6b). Also, there are some minor faults that penetrate deep into the basement, i.e., faults F4 and F7 (Fig. 6a). In addition, the stratigraphic package across the minor faults is roughly tabular. In specific, abrupt changes in the thickness of the stratigraphic package below $T_{70}$ horizon are observed across some minor faults, i.e., faults F4 and F7 (Fig. 6a). In contrast, there are few distinct thickness variations for the stratigraphic package above $T_{70}$ horizon.

In cross-section, minor faults display different assembling styles on the two oppositely-dipping slopes of the Enping sag. On

the northern slope that is controlled by the major fault F1, we observe that minor faults arrange at a regular spacing of c. 2000 m and preferentially dip toward the downdip direction of the major fault F1 (Fig. 8a). This characteristic assemblage style generates a set of domino faults standing on top of the major fault F1. In comparison, on the southern slope that is much gentler, minor faults are either northward or southward dipping, leading to a random distribution of several couples of conjugate faults in the stratigraphic cover (Fig. 8b). Such variation in the assemblage style of minor faults is closely related

to the downdip slip of the gently-dipping major fault F1, which is favourable for newly-formed minor faults to dip toward the slip direction of the underlying major fault.

### 5 Relationship between the low-angle and high-angle normal faults

Based on the fault geometry described above, we note a striking contrast in the dip angle of the rift-related faults in the study area; that is, the rift-related faults can be divided into low-angle (<30° dip) and high-angle (>30° dip) normal faults. Low-

angle normal fault is represented by the major fault F1, whereas most of the other faults are high-angle normal faults, e.g., fault F2. In addition, the high-angle normal faults interact with the low-angle normal fault in various styles, including four interaction styles between them: i) 'isolated fault' that forms independently from and are unaffected by the low-angle normal fault, ii) 'merging faults' that join together at the lower tips, iii) 'abutting fault' that initiates at the low-angle normal fault, and iv) 'cross-cutting fault' that offsets the low-angle normal fault. Key examples of each fault interaction style are

described below in details, focusing on their cross-sectional geometry and interaction manner.





## 5.1 Isolated fault

The first style of fault interaction, i.e., isolated fault, is represented by the relationship between the high-angle fault F7 and the low-angle major fault F1, and is the standard for comparison with other interaction styles. It is observed that the high-angle fault F7 is located far away from and has no interaction with the low-angle major fault F1 (Figs. 6a and 7), suggesting that the major fault F1 plays little effect on the growth of fault F7. Similarly, this style of fault interaction and the evolution course have been described in numerous studies concerning about the interactions between pre-existing and newly-formed normal faults (cf. Duffy et al., 2015; Phillips et al., 2016; Deng et al., 2017b; Fazlikhani et al., 2017).

## 5.2 Merging faults at the lower tips

The relationship between faults F1 and F2 is representative of the second style of fault interactions, i.e., merging faults. In specific, the high-angle fault F2 locates within the hanging wall of the low-angle fault F1 and has a listric shape in cross-sectional view (Fig. 9a). With the depth increasing, the lower part of fault F2 becomes shallower in dip and merges into the plane of fault F1 at the depth of c. 4 s TWT, from where fault F2 shares one common slipping plane with fault F1 toward the southeast. The upper tip of fault F1 terminates at $T_{80}$ horizon and bounds a wedge-shaped growth package between $T_g$ and $T_{80}$ horizons. In addition, the top of the wedge-shaped growth package is partly truncated by erosional unconformity, and is overlain by a tabular stratigraphic package between $T_{80}$ and $T_{70}$ horizons. In contrast, fault F2 bounds a thicker wedge-shaped growth package that comprises the stratigraphy from $T_g$ to $T_{70}$ horizon in its hanging wall. The observations mentioned above indicate that fault F1 started to be active during the deposition of the $T_g$ to $T_{80}$ interval, and became inactive during the deposition of the $T_{80}$ to $T_{70}$ interval. However, fault F2 has been active during the entire syn-rift stage. In other words, faults F1 and F2 initiated during the early stage of the rifting, but fault F2 had a longer active history than fault F1, possibly because of their downward linkage during the late stage of the rifting.

Based on those evidences, the evolution of 'merging faults' at the lower tips is composed of three stages: i) high-angle fault initiates within the hanging wall of active low-angle major fault; ii) high-angle fault becomes gentler as it propagates downwards owing to interaction with low-angle major fault; and iii) high-angle fault merges into the lower part of low-angle fault and continues slipping toward the downdip direction, resulting in abandonment of the upper part of low-angle fault. Such growing history of merging faults resembles the 'merging fault interaction' joining along the margin of shear zone at depth that was described in previous studies about the relationship between intrabasement structure and rift fault (e.g., Phillips et al., 2016; Fazlikhani et al., 2017). They indicated that intrabasement structure acted to perturb the regional stress field and localize strain, causing fault nucleation within its hanging wall, and physically link with the underlying intrabasement structure.





### 5.3 Abutting fault initiating at the low-angle normal fault


The third style of fault interactions, abutting fault, is represented by the relationship between the high-angle fault F4 and the low-angle fault F1. At $T_g$ horizon, fault F4 is situated in the footwall of fault F1, with its eastern tip terminating right at the plane of fault F4 (Fig. 7), which is very similar to the examples of the abutting interaction in previous studies (e.g., Duffy et al., 2015; Deng et al., 2017). In the cross-section traversing the intersection point of faults F1 and F4, we observe that the

low-angle fault F1 tips out at the intersection point, from where fault F4 is subdivided into two segments with different geometries (Fig. 9b). In details, the upper segment of fault F4 is roughly linear at ~60° dips, and bounds a sub-tabular stratigraphic package above $T_{70}$ horizon in the hanging wall. The lower segment of fault F4 existing in the basement displays a listric shape at depth. T-z profile shows that the maximum throw of fault F4, c. 350 ms TWT, is located at the lower tip that intersects with fault F1, from where the fault throw decreases upwards (Fig. 10a). It is unlikely to measure throw data

for the lower segment of fault F4 because there are no discernible reflectors within the basement. Those observations indicate that fault F4 initiated at the upper tip of the low-angle fault F1 and then propagated away from there at a larger dip angle than the low-angle fault F1. Therefore, fault F4 utilized the upper tip of fault F1 as its nucleation site, from where it grew into a high-angle fault and formed the abutting relationship against fault F1.

Another example of abutting fault is represented by the relationship between the domino faults and the major fault F1. In

details, the high-angle domino faults are located within the hanging wall of the low-angle major fault F1, with their lower tips progressively becoming gentler as they approach the plane of fault F1 (Fig. 9c). The lower tip of fault F5, for instance, gradually follows the slipping plane of the major fault F1 with increased depth and finally connects to it toward the downdip direction. Such geometric relationship between faults F1 and F5 is quite similar to the 'mering interaction' between faults F1 and F2. However, they are prominently different in the growth history after having a further examination of fault throw. T-z

profile shows that fault F5 has the maximum throw of c. 150 ms TWT at the intersection point with fault F1, from where the throw value has an upward-decreasing trend (Fig. 10b). Such characteristics indicate that the domino faults initiated at and then propagated upward from the plane of fault F1. We suggest that, during the growth of the domino faults, the underlying fault F1 acted as a basal decollement for the domino faults and fault blocks to slip on top of it with strain accumulation.

### 5.4 Fault cross-cutting the low-angle fault

Based on the examples listed above, it is common that high-angle faults are geometrically and kinematically connected with low-angle major fault. In contrast, there have a few cases that the high-angle and low-angle faults are originally unrelated, and the high-angle fault cross-cuts the low-angle fault afterwards. For instance, the roughly linear and high-angle fault F8 is situated on the gentle slope of the half-graben, with its upper tip extending above $T_{70}$ horizon and its lower tip penetrating into the basement (Fig. 9c). Especially, fault F8 cross-cuts and displaces the trace of the low-angle major fault F1 at the

depth of c. 4 s TWT, generating a vertical offset of c. 200 ms TWT. T-z profile shows that the maximum throw of fault F8, i.e., c. 300 ms TWT, occurs at $T_g$ horizon, from where throw value has a decreasing trend upwards (Figs. 9c and 10c). In





addition, the stratigraphy in the hanging wall of fault F8 is composed of a wedge-shaped growth package between $T_g$ and $T_{80}$ horizons and a tabular package above it. The observations mentioned above suggest that the high-angle fault F8 started to be active during the early stage of the rifting, and experienced a three-step growth history: i) nucleation of high-angle fault
above low-angle fault, ii) fault propagation upward and downward, and iii) ultimate cross-cutting of the underlying low-angle fault. This case shows that high-angle fault initiated above and eventually offset the underlying low-angle fault, which is representative of cross-cutting interaction.

## 6 Discussion

This study uses seismic reflection data to interpret the geometric relationship and evolution of intrabasement and rift-related
structures in the Enping sag, the northern South China Sea. The fault network observed in the study area is characterized by a double-layer structure: the intrabasement thrusting faults at the bottom layer and the rift-related extensional faults in the upper stratigraphic layer. After examining the geometric style and spatial distribution of the intrabasement and rift-related cover structures, we find that the intrabasement thrust faults have different kinematics from the rift-related cover structures, indicating that the former are pre-existing structures formed before the rifting. In addition, we observe that the rift-related
structures are dominated by a low-angle major fault, which interacts with high-angle normal faults in various styles and affects the geometry of neighbouring high-angle new faults. Following discussion will reveal how basement thrust fault evolves and influences the rift-related fault growth and rift system development during the subsequent rifting.

### 6.1 Mechanism of the low-angle normal fault development in the northern South China Sea

Normal fault initiated by an extensional stress field as a simple shear plane in the brittle upper crust is generally at ~60° dips,
which is controlled by the internal and sliding frictions of the rocks (e.g., Anderson, 1951; Byerlee, 1978; Collettini and Sibson, 2001). However, it is not exceptional that very low-angle normal faults, i.e., at or <30° dips, are observed both in the field and rift basins, such as the Mormon Peak Detachment in southern Nevada (Wernicke et al., 1985; Axen et al., 1990; Axen, 1993), Cordillera Detachment Faults in the southwestern US (e.g., Davis and Lister, 1988), and Alto Tiberina Fault in northern Apennines, Italy (Chiaraluce et al., 2007). Similar to those well-known examples in the world, the major fault F1 in
our study area is also a large-scale low-angle normal fault that develops together with high-angle faults in the upper crust during the rifting stage. The mechanics of low-angle normal fault development has been debated for several decades, with the key questions concerning about whether a given fault initiated at shallow dip and whether the fault was active at shallow dip. Previous studies suggested that there are several possible factors that could lead to the occurrence of extensional faults with low dips, mainly including: i) rotation of fault plane from initially high-angle to low-angle (Wernicke and Burchfiel,
1982; Davis 1983; Buck, 1988; Wernicke and Axen, 1988; Axen et al., 1995; Wernicke, 1995), ii) magmatic activity (e.g., Lister and Baldwin 1993; Parsons and Thompson, 1993), iii) elevated pore-fluid pressure or presence of low-friction materials in the fault core (Axen, 1992; Rice, 1992; Hayman et al., 2003; Numelin et al., 2007; Collettini et al., 2009a,





2009b), and iv) reactivation of pre-existing thrust faults (e.g., Coward et al., 1989; Ghisetti and Vezzani, 1999; Collettini et al., 2006; Bird et al., 2015). Following analysis considers the plausible contribution of those factors.

**6.1.1 Late-stage rotation of fault plane from initially high-angle to low-angle**

For the typical rock mechanics following Byerlee's Law, the crust should fail by formation of new high-angle normal faults at ~60° dips, because direct formation of low-angle normal faults at dips <30° is mechanically challenging (e.g., Anderson, 1951; Byerlee, 1978; Collettini and Sibson, 2001). However, low angle normal fault development presents no conflict with Anderson theory if, for example, a fault initiates at ~60° dips and then rotates to low dips <30° while it is inactive. Previous studies have reported several examples of low-angle normal faults that are associated with post-deformational rotation of initially high-angle faults, such as the Snake Range detachment (Miller et al., 1983), Basin and Range faulting in the Yerington district, western Nevada (Proffett, 1977), and Sierra Mazatán core complex in NW Mexico (Wong and Gans, 2008). According to previous studies, the course of fault rotation from high-angle to low-angle can be explained either by the model of a 'rolling hinge' (e.g., Buck, 1988; Wernicke and Axen, 1988; Axen et al., 1995; Wernicke, 1995; Hamilton, 1988), or by late-stage 'domino-style' rotation of normal fault blocks (e.g., Proffett 1977; Wernicke and Burchfiel, 1982; Davis 1983; Wong and Gans, 2008). The 'rolling-hinge' model suggests that flat-lying detachments are rotated from an initial high-angle into a low-angle by upward flexing of the footwall as an isostatic response to unloading (e.g., Buck, 1988; Wernicke and Axen, 1988). The 'domino-style' rotation model, in contrast, suggests that a set of normal faults experience simultaneous rotation of both fault blocks and beds in uniform movement, causing a group of low-angle faults to have the same offset, dip and slip direction. Such a united rotation of faults and fault blocks commonly occurs with the assistance of a weak low-angle layer or structure, i.e., an evaporite layer or a gently dipping detachment fault, at the bottom of the rotated faults with steeper initial dips (e.g., Wernicke & Burehfiel, 1982; Gans & Miller, 1983; Axen, 1988; Wernicke et al., 1985).

Although the models involving post-deformational rotation from high-angle to low-angle fault both geologically and logically make sense, a key issue needs to be solved before discussing whether they fit our case. In specific, the rotation models imply that rotation of initially high-angle fault occurs during or after the initiation of fault plane, so that corresponding footwall rotation and/or stratigraphic truncation is to be observed at the crest of the footwall block. Alternatively, it is likely that a younger half-graben occupies the footwall of the rotated fault owing to relative subsidence created by rotation. In our study area, however, the observations stand in apparent contradiction with the rotation models. Firstly, the top basement in the footwall of fault F1 is sub-horizontal and overlain by a tabular package of post-rift stratigraphy (Fig. 6), which suggests that no new half-graben has formed in the footwall of fault F1, and that there is no demonstrable rotation of the footwall block during and after the deposition of the post-rift stratigraphy. In addition, no other prominent fault with the same orientation and offset develops in the footwall of fault F1, which is unfavourable for simultaneous rotation of fault blocks during fault movement. For the same reason, the flat and sub-tabular syn-rift stratigraphy portrayed in the central part of fault F1 also support the point that the plane of fault F1 undergoes no significant rotation from its active period (Fig. 6b). Secondly, at the southwestern part of fault F1, the hanging-wall syn-rift stratigraphy





is featured by a roll-over structure that expands toward the border fault F1, and gradually thins and flattens on the crest of the hanging wall block (Fig. 6a). This observation represents syn-depositional structure associated with variation in the dip angle of the border fault F1 with depth, i.e., the 'ramp-flat' fault shape, and so these dips may be due to rollover of an independently deforming hanging wall block, rather than a measure of rotation of the fault plane (e.g., Xiao et al., 1991). In
addition, the top basement and overlying syn-rift stratigraphy in the hanging wall of the northern part of fault F1 are prominently tilted toward the border fault, but it is more likely related to a localized warped effect at the interaction zone of faults F1 and F2 (Figs. 6c and 7). Finally, the cross-cutting relationship observed for the high-angle fault F8 and the low-angle fault F1 indicates that fault F1 experienced no strong rotation since it initiated, otherwise fault F8 should be simultaneously rotated to a lower dip (Fig. 9c). Based on the observations mentioned above, there is a small possibility that
the low-angle major fault F1 is an initially high-angle fault that rotates during or after its active period.

### 6.1.2 Magmatic activity

Previous studies have investigated the relationship between magmatism and continental extension, and made a general consensus that magmatism influences mechanisms by which extension is accommodated (e.g., Parsons and Thompson, 1993; Hill et al., 1995; Minor, 1995). Gans et al. (1989) observed that extension in the eastern Great Basin was typically preceded
by a flux of magmatism into the crust and low-angle faulting almost never occurred without accompanying magmatism, indicating that magmatic activity controls the initiation of low-angle faults. Also, previous researches suggested that magmatism could cause a significant heterogeneity in the stress regime that drives low-angle faulting, in the way of either rotating the greatest principal stress orientation off vertical or weakening the crust due to thermal softening at or near magma intrusion (e.g., Parsons and Thompson, 1993; Campbell-Stone et al., 2000). For our study area, thermal events were
suggested to accompany with the onset of the South China Sea rift system, inducing extensive magmatism that led ~30% of the land area to be made up of extension-related plutons and volcanic rocks formed during the Early Cretaceous (Lai et al., 1996; Sewell et al., 2012), and that created the giant NE-trending coastal igneous zone (e.g., Li, 1999; Zhou and Li, 2000). The majority of the NE-trending structures was linked to the intense phase of volcanic and plutonic activity, including the Tolo Channel fault zone system (Sewell et al., 2000). Based on the previous researches mentioned above, it is possible that
magmatic activity played an effect on the formation of the low-angle normal faults during the Late Cretaceous to Early Cenozoic rifting of the South China Sea; however, it is difficult to determine how big the effect is, not to mention that rift-related magmatism is also a characteristic of metamorphic core complex resulting from significant uplift of the footwall of a low-angle fault as described in the 'rolling-hinge' model. Therefore, the effect of magmatism on the local stress field during extension essentially resembles the rotation mechanism of initially high-angle fault as described in the 'rolling-hinge' model,
and more specific examination on the temporal and spatial relationship between magmatism and rift-related structures is needed for determining the mechanism of low-angle fault development.





### 6.1.3 Presence of low-friction materials permitting fault sip with low dips

While models involving post-deformational fault rotation from high-angle to low-angle are supported by numerous field observations, there are also some low-angle extensional faults seem to form or slip with low dips. Examples include the
Whipple-Chemehuevi detachment fault system in the Basin and Range, SE California, which was active at low angles (<30°) based on the evidences of the syn-tectonic strata (Yin and Dunn, 1992). According to previous researches, slip on low-angle normal fault requires some specific circumstances that reduce the frictional strength of the rocks in fault core, including: i) presence of low-friction materials (Hayman et al., 2003; Numelin et al., 2007; Collettini et al., 2009a), ii) elevated pore fluid pressure (Axen, 1992; Rice, 1992), and iii) sub-horizontal weak zone (Yin,1989; Melosh, 1990). Firstly, Lecomte et al.
(2012) investigated the mechanical basis for slipping along low-angle normal faults, and suggested that plastic compaction allows reduction of the effective friction of fault sufficiently for low-angle normal fault to be active at dips of ~20°. More specifically, Haines et al. (2014) collected a series of measurements on the frictional properties of fault gouges from large low-angle normal faults, and indicated that clay gouge formation produces materials with low frictional strength in the fault core, i.e., neoformation of frictionally-weak clay-rich minerals as a result of metasomatic reactions. Those researches
emphasized that low-friction materials associated with some reactions confined to fault zone can act to weaken a pre-existing fault surface, and thus permit it to slip at dips lower than it otherwise could. Secondly, Axen (1992) and Rice (1992) proposed that elevated pore fluid pressure could exert similar effect on decreasing the frictional strength of the rocks, and consequently enable fault to slip at a low dip. They implied that high pore fluid pressure could be focused within fault zone by upper plate mineralization and formation of low-permeability micro-breccia, creating suitable mechanical conditions for
activity on low-angle normal faults. That means it is both mechanically and dynamically favourable for an extensional fault to slip at low dips with the presence of low-frictional mineral or high pore fluid pressure. Thirdly, Fossen et al. (2000) observed that low-angle fault can form along weak layer interbedded within strong layers using plaster experiments. Also, previous studies showed that a basal ductile shear zone could favour low-angle normal fault development by causing stress rotation (Yin, 1989; Melosh, 1990). Such observations indicate that a sub-horizontal weak layer in relatively strong rocks,
i.e., an over-pressured shale, evaporite formation or ductile shear zone, represents an anomalously weak zone permitting extensional fault to flatten along it.

Regarding to our study area, the factor of sub-horizontal weak zone can be firstly excluded, because at the rift onset, the basement rocks were mainly composed of granitic plutons with high frictional strength (e.g., Lu et al., 2011; Yi et al., 2012; Sun et al., 2014). In addition, the depth where the low-angle normal fault existed was not deep enough to be affected by an
underlying ductile shear zone. Thus, the basement rocks should have failed instead by initiation of high-angle normal faults. Since the presence of low-frictional materials, i.e., phyllosilicates, has been found in the clay gouge of the thrust faults in the Pearl River Delta area (e.g., Nanni et a., 2017), we argue that clay gouge formation may have played an effect on the development of low-angle normal faults in the northern South China Sea. However, it is noteworthy that formation of low-friction clay-rich material and elevated pore fluid pressure occur in unusual conditions within fault core, associated with a



low-temperature metasomatic reaction or generation of interior low-permeability zone. As such, fluid-rock diagenesis alters
       the compositions and mechanical properties of fault zone rocks under specific temperature and pressure conditions within the
       fault core, rather than the intact rocks. Therefore, those factors do not directly shed light on the initiation of low-angle
       normal fault in the undeformed rocks, but offer a potential mechanical basis for a pre-existing fault to subsequently move at
       low dips.

### 6.1.4 Reactivation of pre-existing thrust fault

Low-angle normal faults have also been observed in the central Apennies, where their origin is suggested to be subduction
       rollback (Collettini et al., 2006) or collapse of an overthickened accretionary wedge, whereby thrust faults are reactivated as
       low-angle normal faults (Ghisetti and Vezzani, 1999). Also, Ratcliffe et al. (1986) claimed that the border fault of the
       Newark basin in eastern Pennsylvania was a low-angle, at 25°-35° dips, extensional fault resulting from the reactivation of
Palaeozoic imbricate thrust faults in the basement rocks. In addition, evidences of reactivation of Caledonian thrust belts by
       Devonian extensional shear zones were extensively recognized in the northern North Sea, which was thought to be
       responsible for the development of the low-angle normal faults or shear zones at <30° dips in the lower crust (e.g., Coward et
       al., 1989; Bird et al., 2015; Phillips et al., 2016; Fazlikhani et al., 2017). Similar to those researches, we suggest that
       reactivation of basement thrust fault is the primary reason for the formation of low-angle normal fault in our study area on
account of the following proofs.
       The first proof is that we observe a number of pre-existing thrust faults in the basement of the study area, i.e., basement
       faults BF1-BF6. They are separated from the cover structures by the top basement, and are prominently different from cover
       structures in the deformation manner. We therefore interpret them to be originated from the compressional tectonics related
       to the subduction of the Pacific Ocean Plate under the South China Block before the Late Cretaceous to Cenozoic rifting.
This observation is consistent with the study of Ye et al. (2020), which identified several groups of Mesozoic thrust faults in
       the basement of the Pearl River Mouth Basin. In addition, some Early Cretaceous, NE- to E-W-striking thrust faults and NE-
       SW-striking strike-slip faults, e.g., the Tiu Tan Lung Fault and the San Tin Fault, were found in the field outcrops of the
       Pearl River Delta area at the southeast China margin, which are suggested to be of the same origin with those in the offshore
       rift basins (Nanni et al., 2017). Based on the observations from the previous and our studies, we argue that there have
developed pervasive thrust faults in the basement of the study area, which are related to the Late Mesozoic compressional
       event prior to the rifting stage.
       The second and direct proof is that we find some of the rift-related normal faults are explicitly associated with the
       reactivation of basement thrust fault. A good example is the low-angle major fault F1, the along-strike extension line of
       which coincides with the intrabasement fault BF6 (Fig. 6). The overlap between the along-strike extension line of fault F1
and the intrabasement fault BF6 indicates that the low-angle fault F1 initiated by employing a previously existing basement
       fault plane, because pre-existing fault is mechanically weaker than the surrounding basement rocks. In other words, the
       intrabasement fault BF6 and the low-angle fault F1 are two parts of a basement thrust fault before the rifting, and when the





rifting started, one part of the basement thrust fault was employed as the shear plane of the low-angle normal fault F1, whereas the other part remained inactive and was left over to be the intrabasement fault BF6. That is why the intrabasement
fault BF6 overlaps with the extension line of the low-angle fault F1. Another example associated with the reactivation of pre-existing basement thrust fault is the rift fault F3 that shows a gradual decrease in dip angle and merges with the intrabasement fault BF3 with increased depth. The intrabasement fault BF3 has quite similar dips and seismic reflection characteristics with the lower part of the rift fault F3, resembling that they were originally one through-going fault, namely the basement thrust fault BF3. We argue that the lower part of the intrabasement fault BF3 was employed as the shear plane
of the rift fault F3 at the beginning of the rifting, from where the rift fault F3 propagated upward at a steeper dip into the stratigraphic cover. As a result, the upper part of the intrabasement fault BF3 remained inactive, and became a residual short segment situating in the footwall of the rift fault F3. The two listed examples provide a solid proof that the pre-existing basement thrust faults in our study area can reactivate and dominate the development of normal fault system during later rifting stage, and should be responsible for the formation of low-angle normal faults.

In summary, the most unlikely factor for low-angle fault formation in our study area is late-stage rotation of fault plane from an initial high dip to later low dip, because the syn-rift beds are not significantly rotated during and after its active period. To be noted, we cannot rule out that magmatism during rifting is a possible beneficial factor allowing for low-angle fault development because magmatic activity is beyond the focus of our study. We argue that clay gouge formation may play an effect on the development of low-angle normal faults, but cannot be the main factor for the initiation of low-angle fault
because clay gouge is commonly localized within the fault core. From this perspective, we suggest that reactivation of pre-existing thrust fault is ranked the primary factor that controls the formation of the low-angle normal faults in our study area, and possibly with the help of low-friction materials within the fault core. Therefore, our observations support that the low-angle normal fault bounding the Enping sag was originated from the reactivation of a basement thrust fault forming before the rifting, which is consistent with the suggestion of Ye et al. (2018).

**6.2 Reactivation mode of basement thrust fault and controlling factors**

Theoretically, pre-existing weak fault or shear zone originated from an early tectonic event is prone to be reactivated during a second tectonic stage. Actually, reactivation of pre-existing fault or shear zone and resultant interactions with rift-related normal faults have been reported in many rifts that experienced multiple phases of rifting, e.g., the northern North Sea (e.g., Badley et al., 1988; Færseth, 1996; Færseth et al., 1997; Odinsen et al., 2000; Whipp et al., 2014; Duffy et al., 2015; Deng et
al., 2017a; Fazlikhani et al., 2017), Gulf of Thailand (e.g., Morley et al., 2004, 2007), and East African Rift (Corti, 2009; Le Turdu et al., 1999; Lezzar et al., 2002; Muirhead & Kattenhorn, 2017). For example, Duffy et al. (2015) carried out a detailed study in three-dimensional geometries of the reactivated first-phase faults and interaction styles with the second-phase faults in the northern North Sea, with the main goal of investigating the fault network evolution in multiphase rifts. Also, Morley et al. (2004) suggested that Palaeozoic and Mesozoic orogens exerted a strong influence on the development of
both Tertiary strike-slip and normal faults in the Gulf of Thailand. As mentioned above, the development of the low-angle





fault F1 is considered as the result of reactivation of a basement thrust fault, implying that the fault evolution in our study area is influenced by reactivation of pre-existing basement faults.

Our focus here is to figure out the varieties in the reactivation pattern of basement thrust fault. For instance, our observations show that the reactivation of the basement thrust fault BF6 is partial along strike; that is, one part of the basement thrust fault

reactivates and generates the low-angle fault F1, while the other part remains as an intrabasement fault being buried by later rift sequences (Fig. 6d). In addition, the low-angle fault F1 accumulates more strain than neighbouring normal faults, growing into the major fault with a bigger fault length and displacement. Another example of partial reactivation is represented by the basement thrust fault BF3 that is bisected by the rift fault F3 in cross-section view. In details, the lower part of the pre-existing basement fault BF3 unites with the newly-formed fault F3 and slips together as one through-going

fault, whereas its upper part lies in the footwall of fault F3 and has no evidence of down-dip slip during the rifting stage. Those examples suggest that the reactivation of basement thrust fault is not simply reutilizing the pre-existing plane of thrust fault and reversing the slip direction of fault block, but is characterized by partial reactivation in cross-section or map view, causing the remnants of thrust fault to be left within the basement. Besides, we note that a lot of intrabasement thrust faults have no activity during the rifting stage. For instance, the intrabasement faults in the hanging wall of major fault F1, i.e.,

BF1, tips-out at the top basement, which is a sign of no movement after when the rift sequences overlie it (Fig. 5a). Similarly, there are also some intrabasement faults occurring in the footwall of fault F4 that are representative of no activity during the rifting stage as well, because their upper tip terminates below the top basement (Fig. 5c). More cases of unactivated basement thrust faults have been found in the onshore area of the northern South China Sea, where outcrops of basement thrust faults was widely reported (Nanni et al., 2017). Even though full reactivation is not directly observed here, it is likely

to occur to the south of the study area, e.g., the Baiyun and Liwan sags, because of a higher potential for pre-existing structures to be reactivated at the proximity of rift centre where the crust could be hyperextended (Wang et al., 2018). Besides, full reactivation of pre-existing fault has been well portrayed in other natural rifts, e.g., the Horda Platform in the northern North Sea (e.g., Bell et al., 2014; Duffy et al., 2015; Phillips et al., 2016; Fazlikhani, et al., 2017), and numerical models investigating multiple phases of rifting (e.g., Deng et al., 2018), from which we can learn lessons.

Based on the observations from the northern South China Sea (e.g., Nanni et al., 2017; Ye et al., 2018, 2020; Zhou et al., 2019), other natural rift basins (e.g., Morley, 2004, 2007; Bell et al., 2014; Duffy et al., 2015; Phillips et al., 2016; Fazlikhani et al., 2017), and numerical models (e.g., Deng et al., 2017b, 2018), the reactivation of a basement thrust fault can be classified into three modes: i) no reactivation, ii) partial reactivation, and iii) full reactivation. For a better understanding of the variety in the reactivation mode of basement thrust fault, we develop a conceptual model highlighting the key

characteristics of reactivation pattern of basement thrust fault and interaction style with adjacent normal faults (Fig. 11). In details, unactivated intrabasement thrust fault will keep as blind fault beneath the top basement and is to be cross-cut by newly-formed high-angle faults, which provides a solid evidence for the presence of pre-existing basement structures prior to rifting (Fig. 11a). In contrast, partly or fully reactivated basement thrust fault is able to develop into low-angle normal fault under favourable conditions during later rifting stage, and influence the geometry of newly-formed high-angle faults. For



example, a partly reactivated basement thrust fault can slip at low dips and become a major fault, with its inactive portion being preserved as a hint of partial reactivation of pre-existing fault. In addition, the reactivated portion of the basement thrust fault is featured by merging and abutting interactions with newly-formed high-angle normal faults (Fig. 11b). Furthermore, a fully reactivated basement thrust fault has similar fault interactions with partly reactivated one, i.e., merging and abutting interactions; however, it is prominent that the resultant fault network appears geometrically different (Fig. 11c).

In details, fully reactivated basement thrust fault has a through-going fault trace with a bigger length and displacement, whereas partly reactivated one consists of a few numbers of separated fault segments with smaller lengths and displacements. As a whole, the fault network comprising reactivated basement thrust faults and newly-formed normal faults become more complex with an increasing reactivation extent, because of a larger possibility for fault interactions with each other. This conceptual model can be largely applicable to the rift setting that is influenced by pre-existing basement thrust faults.

We also observe that unactivated basement thrust faults are much more pervasive than reactivated ones, because numerous unactivated basement thrust faults are found in the hanging-wall and footwall blocks of the major fault F1 (Figs. 5 and 6). That means, reactivation of the basement thrust faults is selective, with just a small proportion of them preferring reactivation. For the purpose of understanding the factors controlling the reactivation of pre-existing structure, physical analogue and numerical models have been employed to investigate the reactivation pattern of pre-existing fault and the

dynamic mechanism under an extensional background. According to previous studies, reactivation of a pre-existing fault mainly depends on fault strength relative to the surrounding rocks (e.g., Bellahsen & Daniel, 2005; Dubois et al., 2002; Etheridge, 1986; Ranalli & Yin, 1990), state of stress in three dimensions (e.g., Ranalli & Yin, 1990; Sibson, 1985), strain magnitude (e.g., Henza et al., 2010, 2011), and extension direction (e.g., Bonini et al., 1997; Deng et al., 2017b, 2018; Fazlikhani et al., 2017; Henza et al., 2010, 2011; Keep & McClay, 1997; Phillips et al., 2016). Regarding to our case, the

different reactivation modes of basement thrust faults are not likely to be related to different fault strength, because the basement thrust faults are of same age and origin from the Middle Jurassic to Early Cretaceous compressional event, so that they should have similar strength. Henza et al. (2010, 2011) used scaled analogue models to investigate how extension direction and strain magnitude influence the reactivation pattern of first-phase faults during later extension. They observed that strain magnitude could control the density and length of faults of each extensional phase, and so dominated the

geometries of the final fault network. They also suggested that reactivation of first-phase faults is to be expected for angles as high as 45° between the two extension directions, and the reactivation has a positive correlation with the angle between the extension directions. Such variety in the reactivation mode coexists where pre-existing faults have various orientations (e.g., Claringbould et al., 2017), but in our case there is no direct correlation between fault orientation and reactivation mode, because unactivated and reactivated NE-SW-striking basement thrust faults are both observed. Since strain magnitude in one

rift system increases from rift margin to rift centre, we suggest that the reactivation potential of basement thrust faults increases in that direction as well. In addition, Kirkpatrick et al. (2013) suggest that selective reactivation of pre-existing structures is greatly dependent on their scale and dip, with large-scale, steeply-dipping structures preferentially reactivated and smaller, shallowly-dipping structures cross-cut. Another possible reason is the strain shadow zone encircling reactivated





thrust faults. We argue that the strain is likely to be localized on a few reactivated major faults, i.e., the major fault F1 that
has a big length and displacement, causing strain shadow zone to form in their vicinity. If so, other small basement faults in
the strain shadow zone are prevented from reactivation. This is consistent with the suggestion made by Ackermann and
Schlishce (1997) that a stress-reduction shadow naturally forms around active master faults, where the nucleation of smaller
faults is retarded. Therefore, the combination of overall strain distribution, scale of pre-existing faults and strain shadow
zone surrounding reactivated ones leads to a limited number of fault reactivation in rift basin underlain by pervasive
basement thrust faults.

**6.3 Impacts of basement thrust fault on normal fault development**

According to previous studies on fault evolution of multiphase rift, the impact of pre-existing structure or fault on the
geometry and evolution of normal fault mainly involves altering fault density (e.g., Willemse et al., 1996; Ackermann and
Schlische, 1997; Gupta and Scholz, 2000; Cowie and Roberts, 2001; Soliva et al., 2006), strike and dip (e.g., Henza et al.,
2010, 2011; Deng et al., 2017b, 2018; Morley et al. 2004, 2007), displacement (e.g., Duffy et al., 2015; Deng et al., 2017b,
2018), and offering nucleation sites to neighbouring new faults (Morley et al., 2004, 2007; Phillips et al., 2016; Deng et al.,
2017b). Similar to multiphase rifts, the fault development in our study area is also greatly influenced by the reactivation of
pre-existing basement faults, and is characterized by the formation of non-collinear faults and various styles of fault
interaction between reactivated basement faults and newly-formed faults. For instance, fault density is relatively low in the
hanging wall of the major fault F1, implying the presence of strain shadow zone around fault F1 during movement. In
addition, the geometry of faults F2 and F5 reflects the influence of reactivated fault on the dip and displacement of new
faults. However, the particularity of our study area is that the low-angle normal fault F1 associated with the reactivation of
basement thrust fault developed at broadly the same time and location with the high-angle ones, and played a dominant role
in the fault evolution during the rifting. Such importance of low-angle fault on the normal fault growth and rift development
has also been emphasized in the Basin and Range (e.g., Axen et al., 1990; Campbell-Stone et al., 2000; Hamilton, 1988) and
the northern North Sea Rift (e.g., Phillips et al., 2016; Fazlikhani et al., 2017). Now we focus on how the reactivation of
basement thrust fault affects the growth of newly-formed high-angle faults.

According to the aforementioned observations, the first key impact is that reactivated basement thrust fault affects the dip
angle of overlying rift faults, represented by the 'merging fault' that initiates at high dips within the hanging wall of the low-
angle major fault F1 and has an obvious decrease in dip as it propagates downward the slipping plane of fault F1 (Fig. 9a).
Such effect indicates that the dip of rift-related faults may be influenced by the dip of the underlying low-angle fault. This
observation can be explained by the anisotropic mechanics of the rock adjacent to the major fault F1; that is, the major fault
F1 underwent a long-term activity from the Mesozoic thrust fault to the Cenozoic extensional fault, generating a deformation
zone full of small-scale interweaved fractures and/or low-friction materials in the fault rocks that are weaker than the
surrounding undeformed rocks (e.g., Scheiber et al., 2015). As such, normal fault is likely to slip at a low dip given that the
rocks are sufficiently weakened. In addition, this weakening effect lays a foundation for the downdip amalgamation of faults





F1 and F2, which results in the abandonment of the upper part of fault F1 after the linkage. This effect of basement structure on the dip of later rift-related faults is consistent with the suggestion made by Ring (1994) that Proterozoic basement structures represented the basic anisotropy influencing pre-Cenozoic rifts and also the Cenozoic Malawi rift, and echoes that

low-friction materials forming within fault zone can act to weaken a pre-existing fault surface, and thus permit it to slip at lower dips (Haines et al., 2014).

Another key impact is that reactivated basement thrust fault offers nucleation sites for newly-formed normal faults. For instance, the coincidence between the position of the throw maximum of fault F4 and its intersection point with the upper tip of fault F1 suggests that fault F4 initiates at the upper tip of the low-angle fault F1, from where it propagates radially as a

new high-angle fault plane (Fig. 9b). In addition, the domino faults developing in the hanging wall of the low-angle fault F1 have similar growth pattern. The newly-formed domino faults nucleate at the plane of the low-angle fault F1, and then propagate upwards from it at higher dips, indicated by the throw maximum of the domino faults occurring at the intersection with the low-angle fault F1(Fig. 9c). This impact on the nucleation site of newly formed faults can also be explained by the local weaknesses/fractures within the deformation zone of the major fault F1. Based on the two mentioned examples, we

argue that there will be strain localization on these weaknesses/fractures during the extension and that they are prone to be employed as nucleation sites of future faults. Phillips et al. (2016) and Fazlikhani et al. (2017) described similar fault growth pattern for fault nucleation on low-angle shear zone in the northern North Sea, where newly-formed faults nucleated within basement shear zone and gradually propagated away from it. They suggested that basement shear zone acts as pre-existing weakness for providing nucleation sites of future faults, which is consistent with our observation.

The above two impacts are suggestive of two different evolution models of a reactivated pre-existing fault and nearby new faults during rifting: i) the 'decoupled model' that reactivated pre-existing fault and nearby new faults are initially isolated segments at different levels, followed by a later stage of vertical propagation and final linkage to form a connect fault system (Fig. 12a; Childs et al., 1996; Schöpfer et al., 2006; Jackson & Rotevatn, 2013; Deng et al., 2018); and ii) the 'coupled model' that new faults nucleate at pre-existing fault and grow as kinematically related components of a fault array, where pre-

existing fault serves as nucleation sites of new faults (Fig. 12b; e.g., Deng et al., 2018; Duffy et al., 2015; Henza et al., 2010, 2011; Baudon and Cartwright, 2008; Morley et al., 2004; Walsh et al., 2002, 2003). For the case of faults F1 and F2, we find that fault F2 evolves from an initially isolated fault segment in the hanging wall of the low-angle major fault F1 to ultimate merging into fault F1 and formation of a continuous fault system, which resembles the growth history of 'decoupled model'. Such decoupling evolution of reactivated pre-existing fault and neighbouring new faults is described in three-dimensional

numerical models, which indicates that new faults are capable to initiate independently in the succession overlying the pre-existing fault if the overlying succession is thick enough (e.g., Deng et al., 2018). As for our case, the reason that the reactivated pre-existing fault and new faults develop an initially decoupled pattern is possibly related to the thick thrust belts existing in the upper part of the basement before rifting, so that new rift-related faults will need to accumulate sufficient deformation for cutting off the thick basement rocks before reaching the reactivated fault. In comparison, the case of the

domino faults that nucleate at the reactivated basement fault F1 has similar features with the 'coupled model', because they





are both geometrically and kinematically linked to the major fault F1 in their early history and then grow upward as a connected fault system with strain accumulation. Childs et al. (1995, 1996) suggested that individual fault segments initiate and grow as kinematically related components of a fault array, becoming hard-linked to a connected fault system over time in a coherent fault model, which is similar to the 'coupled model' of the evolution of reactivated basement thrust fault and
domino faults. The 'coupled model' possesses the general notion that pre-existing fault is a weakness being favourable for nucleation of future faults. We suggest that the reason for the coupled evolution of the reactivated basement fault and domino faults is that the syn-rift stratigraphic cover where the domino faults develop is much thinner and weaker than the underlying basement rock, so that the newly-formed domino faults can cut through the syn-rift stratigraphy with a small amount of strain accumulation. Summarizing, the two different models of normal fault evolution affected by reactivation of
basement thrust fault imply that reactivated basement thrust fault exerts a significant effect on the geometry and growth pattern of new faults within the hanging wall, which are the key impacts of pre-existing thrusts on normal fault development. In addition, the difference in the evolution models reflects that the growth pattern of a reactivated pre-existing fault and nearby new faults is closely determined by rock strength of the hanging-wall block. Based on the above analysis, we provide a preliminary case study for identifying the relationship between pre-existing thrust faults and new rift faults and improving
the understanding of fault evolution in rift basin affected by pre-existing thrusting structures, which has been paid less attention in previous researches.

**6.4 Implications for tectonic evolution of the northern South China Sea**

A variety of models have been proposed for explicating the opening mechanism of the South China Sea, but the most famous models are two types: i) the 'pull-apart model' suggesting that the opening is attributed to southeast extrusion of the
Indochina Block driven by the collision between India and Asia, and accompanied with the formation of a number of pull-apart basins at the margins of the South China Sea (Briais et al., 1993; Gilley et al., 2003; Leloup et al., 1995, 2001; Tapponnier et al., 1982, 1990, 2001); and ii) the 'slab-pull model' indicating that the opening results from slab rollback and retreat of the paleo-Pacific Plate linked with northward subduction of a proto-South China Sea at the north Borneo Trench, which leads to back-arc extension, rifting and seafloor spreading of the South China Sea (Hall, 2002; Lee and Lawver, 1995;
Morley, 2002, 2012; Rangin et al., 1995; Taylor and Hayes, 1980, 1983). Both models raise a general hypothesis that the South China Sea underwent a transition from convergent Andean-type continental margin to divergent Western Pacific-type margin during the Late Mesozoic (Holloway, 1982; Taylor and Hayes, 1983; Li and Li, 2007; Li et al., 2012; Shi and Li, 2012). As analysed above, our study supports that the South China Sea rift system is built on the inhomogeneous basement comprising a series of pre-existing thrust faults that stem from an earlier compression/subduction tectonics and exert a
significant role on the subsequent rift evolution and normal fault development. Apart from our study area, the neighbouring sags within the Pearl River Mouth Basin and adjacent rift basins, e.g., the Zhu I Depression (Ye et al., 2020), Chaoshan Depression (Li et al., 2008; P. Yan et al., 2014), the Qingdongnan basin (Hu et al., 2013), the Taixinan basin (Li et al., 2007), and even the Nansha Trough at the southern margin of the South China Sea (Yan and Liu, 2004; Wang et al., 2016), also





contain a number of pre-existing NE-SW- to NEE-SWW-trending and nearly E-W-trending basement fabrics that are
believed to affect the development of the rift-related faults and basins during the Late Cretaceous to Cenozoic rifting (Hu et
al., 2013; Lister et al., 1986; Sun et al., 2009, 2010; Zhou et al., 1995; Zhu and Jiang, 1998; Ye et al., 2018, 2020). In
addition, some NE-SW- and E-W-trending thrust faults and strike-slip faults of the Jurassic to Early Cretaceous have been
found in the field outcrops of the Pearl River Delta area, at the northern margin of the South China Sea, and some of them
were reactivated by Cenozoic extensional structures (Nanni et al., 2017). Based on the above-mentioned researches, we
agree that a Mesozoic subduction zone have developed along the northern margin of the South China Sea (e.g., Li et al.,
1999; Zhou et al., 2006, 2008; Min et al., 2010; Xu et al., 2016), or further south along the Dangerous Ground and North
Palawan (e.g., Morley, 2012; Yan et al., 2014), which is thought to be associated with the Jurassic-Cretaceous Yanshanian
movement due to subduction of the paleo-Pacific Plate beneath the South China Block (Faure et al., 1996; Ren et al., 2002;
Shu et al., 2006; Zhou et al., 2006). Our study is consistent with previous studies about the origin of the basement thrust
faults from a Mesozoic subduction zone, and supports the view that the formation of the South China Sea rift system and the
reverse reactivation of the basement thrust faults are mainly governed by the dynamic transition from compressional to
extensional tectonic setting. Therefore, we argue that the tectonic evolution of the northern South China Sea rift, especially
the area close to the Mesozoic subduction zone, should be greatly influenced by quite a few groups of basement-rooted
thrusting structures, and so examining their spatial and temporal evolution is of vital importance for understanding the
structural styles of the subsequent rift and fault system.

Integrating the observations from our study and neighbouring rifting areas, we construct a conceptual model illustrating how
the presence of intrabasement thrust faults influences the geometry and evolution of subsequent rift system (Fig. 13). Firstly,
reactivated basement thrust fault controls the location and geometry of rift boundary fault (Fig. 13b-c). In specific, fault F1 is
suggested to be resulted from the reactivation of a pre-existing basement thrust fault, and later grows into the low-angle
major fault bounding a wide and shallow half-graben in the hanging wall (Fig. 6). This point is consistent with Walsh et al.
(2002) and Deng et al. (2018), which proposed that reactivation of pre-existing fault is characterized by 'near-constant'
growth model that reactivated basement fault rapidly obtains its near-maximum length and becomes the major fault,
followed by a long period of displacement accrual. In addition, the low-angle major fault F1 bounds a wider syn-rift
subsidence area in the hanging wall owing to its gentler dip relative to high-angle faults. As such, strain is distributed over a
much wider and shallower basin area on top of the hanging-wall block of the low-angle boundary fault, leading to a more
gently-dipping landform from rift margin to depocenter (Fig. 13b). We therefore argue that rift system development affected
by underlying basement thrust faults will generate specific fault geometry, basin topography and depositional system that
differ from those established models of rift evolution, because syn-rift stratigraphic sequence is sensitive to a change in the
basin paleotopography (e.g., Gawthorpe & Leeder, 2000; Cowie et al., 2006). This point will play an important role in
subsequent investigation of basin sedimentology and petroleum geology, so bearing it in mind is essential for geologists to
predict sediment infilling pattern, facies and hydrocarbon reservoirs in those rifted basins. Secondly, there will be a
decreasing trend in the reactivation potential of basement thrust faults from rift centre to rift margin, due to a decrease in the





strain magnitude toward rift margin (e.g., Cowie et al., 2015). In details, without considering other aspects, there will be more chances for reactivation of basement thrust faults near rift centre, whereas at rift margin, basement thrust faults are

prone to remain unactivated (Fig. 13b). Such a scenario provides a template for future fault and basin structure development in rift affected by underlying basement structures. Nearby rift centre, for instance, newly-formed faults either develop as minor structures in the space between major reactivated basement faults in order to adjust interior deformation of fault blocks, or merge into the slipping plane of reactivated faults to form a connected fault system (Fig. 13c). In comparison, at the rift margin where basement thrust faults hardly reactivate and affect new fault development, the fault network should be

featured by a group of evenly spaced, sub-parallel new faults offsetting pre-existing basement thrust faults, which is similar to the rift development within relatively homogeneous crust (e.g., Gupta et al., 1998; Cowie et al., 2000; Gawthorpe and Leeder, 2000). Summarizing, we argue that the tectonic evolution of the northern South China Sea is greatly influenced by the presence of intrabasement thrust faults, in the way of controlling the location and degree of faulting that occurs during rifting, and thus the overall basin structures and paleotopography.

**7 Conclusions**

This study aims to investigate how basement thrust fault evolves and influences the development of rift system and normal fault during subsequent rifting, and what enlightenments that influence has on the subsequent basin structure and syn-rift stratigraphic sequence. Our observations suggest that basement thrust fault could reactivate and slip as low-angle normal fault during later rifting, and then exert significant effects on the nucleation, dip and displacement of nearby new faults,

resulting in several styles of fault interactions, e.g., merging, abutting and crosscutting. Key conclusions obtained from this investigation include:

(1) Basement thrust faults were pervasively distributed in the South China Sea area before the Late Cretaceous to Early Cenozoic rifting phase, and their later reactivation is the primary cause for the development of low-angle normal fault during rifting, possibly with the assistance of low-friction materials in the fault core.

(2) Reactivation mode of basement thrust fault includes full, partial and no reactivation, depending on overall strain distribution across rift, scale of basement thrust fault and strain shadow zone surrounding reactivated ones.

(3) Reactivated basement thrust fault influences the nucleation, dip and displacement of nearby new faults, causing them to nucleate at or merge into the fault plane downwards.

(4) Rift structure affected by underlying basement thrust structures are characterized by shallowly-dipping basin topography,

and formation of shallower and wider basin depocenter for syn-rift sediment infill, which is crucial for further investigation of basin sedimentology and petroleum geology.

(5) Tectonic evolution of the northern South China Sea rift, especially the area close to the Mesozoic subduction zone, is greatly affected by quite a few groups of basement-rooted thrusting structures, and so examining their spatial and temporal evolution is of vital importance for understanding the structural styles of the subsequent rift and fault system.



**Acknowledgements**

This study is supported by Shenzhen Branch of CNOOC of China (Project No. 2017-RFPSZ-0613), China Postdoctoral Science Foundation (No. 234012000002), research fund from the State Key Laboratory of Continental Dynamics and Department of Geology, Northwest University (No. 111110005) and National Natural Science Foundation of China (No. 42002124). We appreciate the thoughtful discussions and suggestions from Ge zhiyuan at China University of Petroleum

(Beijing), and thank Wang tianbao for his support and assistance in data collection.

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



Fig. 1 Location and simplified structural map of the study area. (a) Topography and bathymetry of the South China Sea (SCS) and adjacent regions. (b) Map of structural divisions of the Pearl River Mouth Basin and location of the Enping Sag. EPS—Enping Sag; XJS—Xijiang Sag; XNS—Xingning Sag; HZS—Huizhou Sag; LFS—Lufeng Sag; HJS—Hanjiang Sag; BYS—Baiyun Sag; LWS—Liwan Sag; XNS—Xingning Sag. (c) Schematic structural map and depocenter of the Enping sag at the basement. Straight lines show the location of cross-sections. (d) Transect across the middle part of the Pearl River Mouth Basin, showing the geological interpretation of the structure of the northern SCS margin. After Yang et al.(2018); Ye et al.(2018).





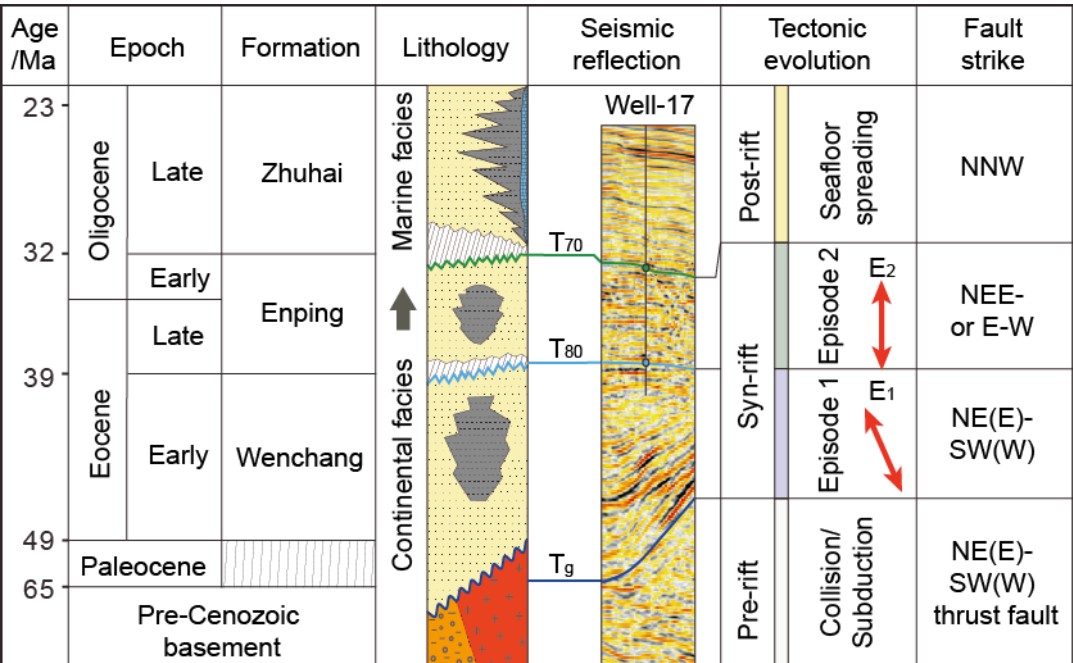

**Fig. 2 Tectono-Stratigraphic evolution column of the Pearl River Mouth Basin, showing the geological correlation with seismic section and the key seismic horizons interpreted throughout this study, along with the major tectonic events to have affected the region. After Ye et al. (2018). Colours of horizons and stratigraphic intervals are consistent and referred to throughout the text.**
5 **Red arrow represents the extension direction of each rift episode.**

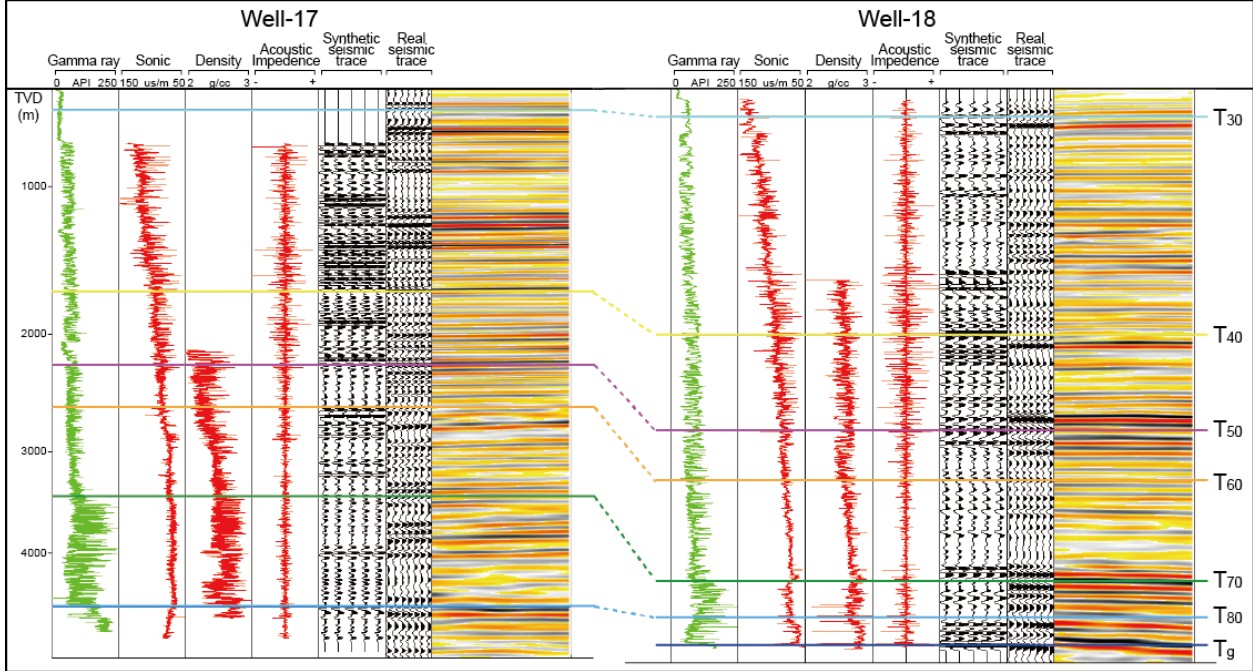

**Fig. 3 Seismic-well ties for Well-17 and Well-18. See Fig. 1c for the location of the wells. Seismic data are Normal Polarity (SEG Convention), whereby a peak (brown reflection) represents an increase in acoustic impedance and a trough (black reflection) a**





decrease in acoustic impedance. $T_{30}$ to $T_{60}$ are correlated seismic horizons in the Pearl River Mouth Basin, and $T_{70}$ to $T_g$ are the
10  key seismic horizons interpreted in the study area.

**Fig. 4 Methods of interpreting intrabasement and rift-related structures. (a) Cross-section showing the observed fault structures in the stratigraphic cover and basement rocks. (b) A close-up showing the observed thin intrabasement reflection packets, and (c) A close-up showing the interpretating method of rift-related fault structure.**


**Fig. 5** Uninterpreted (Left panels) and interpreted (Right panels) cross-sections showing the occurrence of intrabasement structures in the study area. (a) and (b) show the characteristics of Type I intrabasement structure, while (c) shows the interpretation of Type II intrabasement structure with Type I intrabasement structure. Question mark indicates uncertainty of the interpretation. See Fig. 1 for location. Colours within the sedimentary cover correspond to ages shown within the stratigraphic column (Fig. 2).



Fig. 6 Cross-sections perpendicular to the major structural trend showing the geometry of the major fault F1 and hanging-wall half-graben. (a)-(d) are four uninterpreted and interpreted cross-sections sub-perpendicular to the local strike across fault F1, with the simplified interpretation of stratigraphy. The colours used for the key horizons and stratigraphic intervals between them are consistent with those used in Fig. 2. See Figs. 1 for location.









**Fig. 7** Upper panel is the TWT structure map of the top basement showing the location of key fault traces and sub-sags. Color bar shows the depth of the top basement from the ground. Red is the uplift, purple is the depocenter. Lower panel shows the T-x plots of the major fault F1.





**Fig. 8 Cross-sections showing the assemblage style of minor faults. (a) Domino faults developing on the slipping plane of the major fault F1. (b) Conjugate fault developing on the gentle slope of the Enping sag. See Figs. 1 for location.**





Fig. 9 Cross-sections showing interaction styles between the low-angle major fault and high-angle faults found in the Enping sag. (a) 'Merging faults' that join together at the lower tips (F1 and F2). (b) 'Abutting fault' that initiates at the low-angle normal fault (F1 and F4). (c) 'Abutting fault' (F1 and F5) and (d) 'Cross-cutting fault' that offsets the low-angle normal fault (F1 and F8). See Figs. 1 for location.





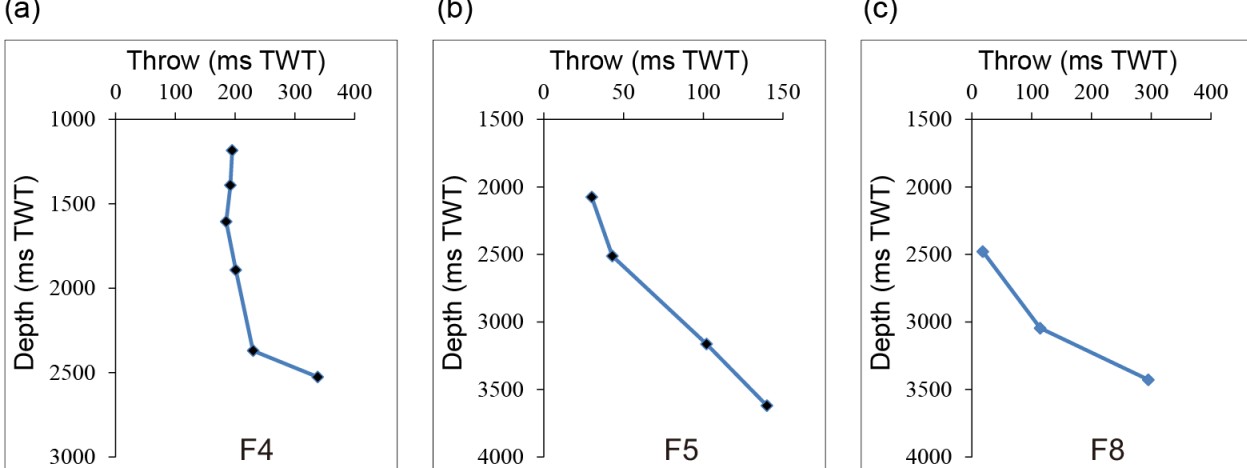

**Fig. 10 T-z profiles of fault F4, F5 and F8.**

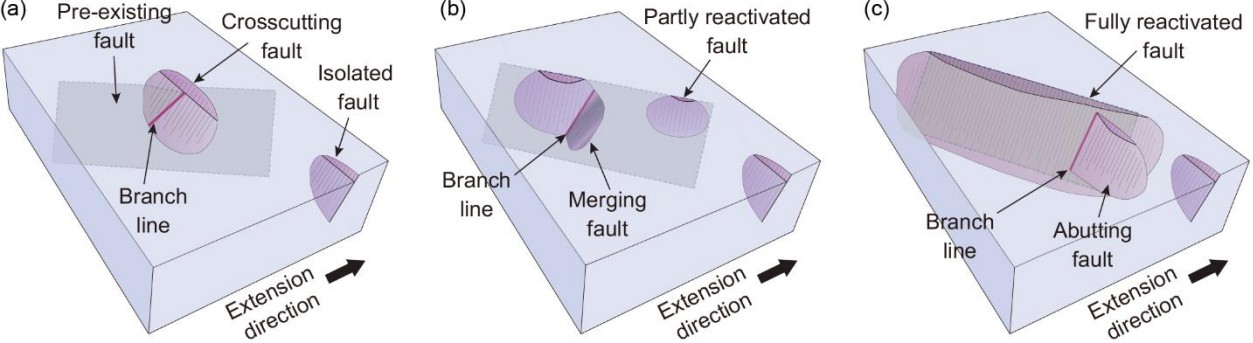

5    **Fig. 11 Summary figure showing the variety in the reactivation mode of pre-existing thrust fault and fault interaction styles developing during later rifting. (a) No reactivation. (b) Partial reactivation. (d) Full reactivation. Grey rectangle is pre-existing thrust fault existing beneath the surface. Purple oval is new fault. Black fault traces indicate the surface expressions of faults. Purple line is the branch line at the intersection between fault pairs.**



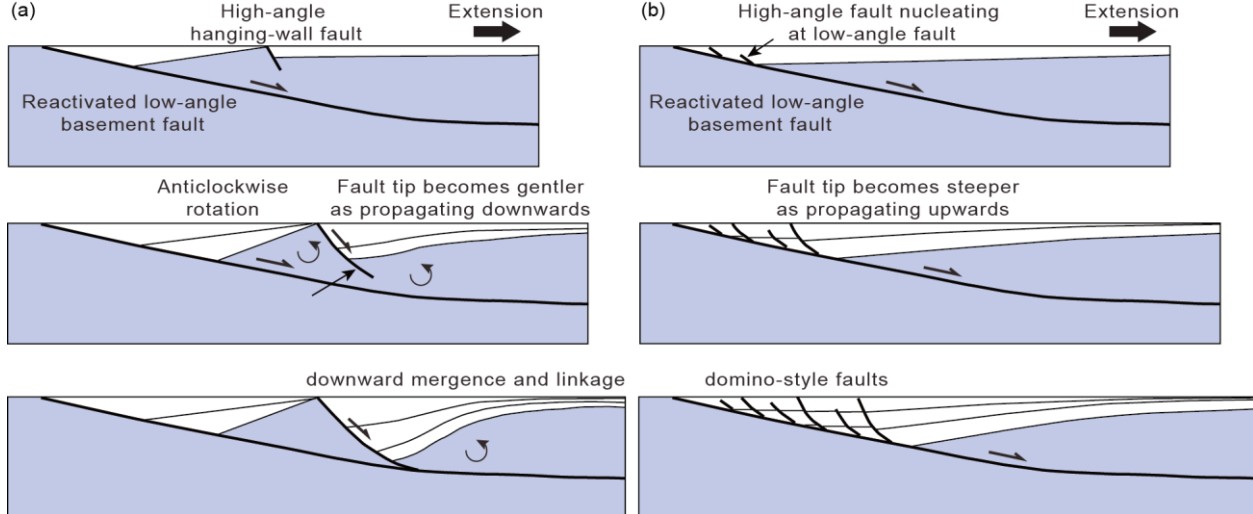

10 **Fig. 12 Model of fault evolution between reactivated basement thrust and newly-formed high-angle normal faults during rifting. (a) 'Decoupled model' that that reactivated thrust fault and nearby new faults are initially isolated, followed by a later stage of vertical propagation and final linkage to form a connect fault system (b) 'Coupled model' that new faults nucleate at reactivated fault and grow as kinematically related components of a fault array, where the reactivated thrust fault serves as nucleation sites of new faults.**



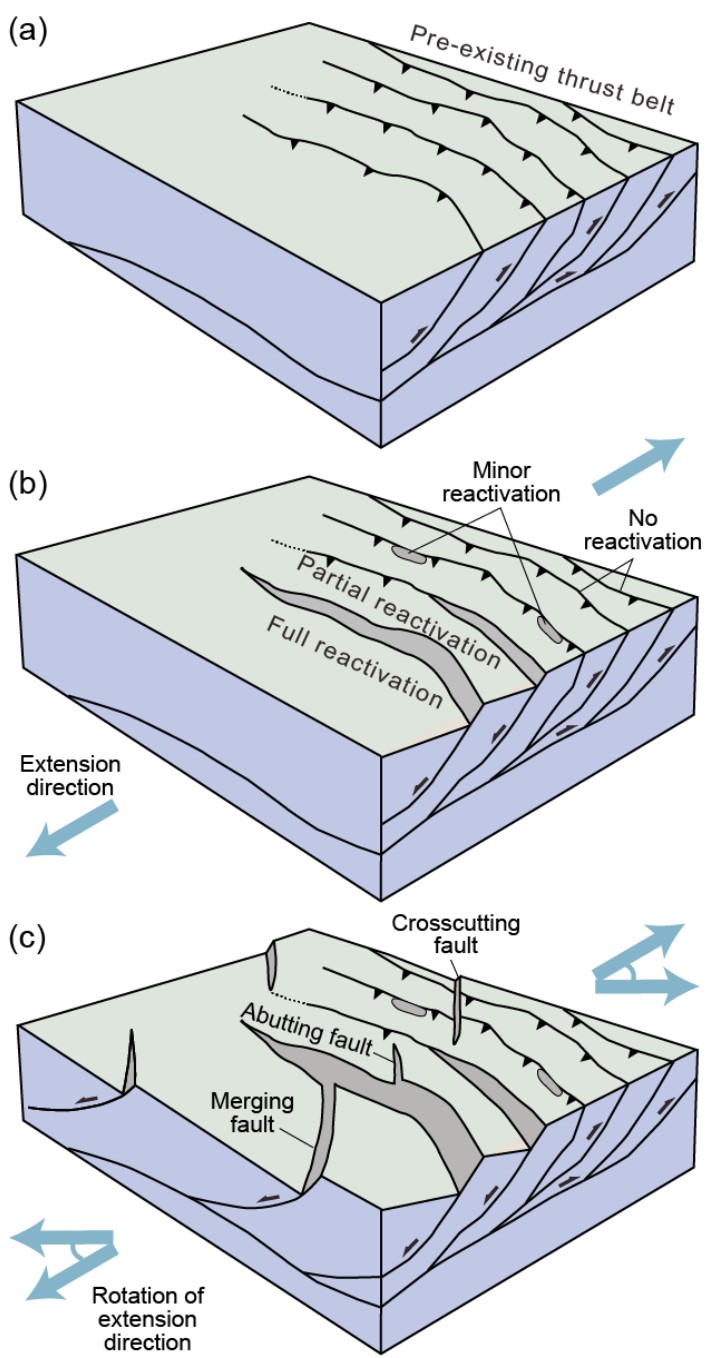

**Fig. 13 Synoptic figure showing how the presence of basement thrusting structures may influence the geometry and evolution of overlying rift system. (a) Pre-rift framework of thrusting structures within crystalline basement. (b) Upon orthogonal extension those pre-rift thrusts act to control the reactivation extent, location and degree of faulting, and basin-boundary fault that occurs during rifting. (c) Subsequent rotation of extension direction produces a range of interactions with the rift-related faults, affecting the overall basin structures and paleotopography.**

