# Peer review of "Impact of basement thrust fault on low-angle normal fault and rift basin evolution: a case study in the Enping sag, Pearl River Mouth Basin"

_Solid Earth, 2021_

## Referee Comment (RC2)

[referee-annotated manuscript omitted]

---

## Author Response (AR1)

Review 1

Below are some major and minor comments:

1. The only major problem in this manuscript is the lengthy description and discussion. Some parts are repeated and unneeded. For example, the section 6.1 has 150 lines to discuss the possible model for the low angle faults. These models are well-known in interpretating the angle of faults, but several sentences are sufficient for readers to understand the reason why the formation model is adopted. Too many words may have countereffects. I thus recommend shortening this section, leaving section 6.1 without any subsections, and briefing the key evidence for the inherited tectonic model. This kind of problem also applies in the structural description in section 5, and the rest parts of discussion. The current discussion is too long to get the key ideas.

   We skim through the manuscript and brief the text in sections 5 and 6 now. In specific, we shorten the description of the literatures but preserve their main ideas in the discussion. And we delete some sentences in section 5 to make it more concise.

2. The abstract could be improved with some more implications. The discussion gives them on the evolution and interaction between the thrust faults and normal faults, they should be introduced at the end of it.

   More implications are added in the abstract. See Lines 20-25.

3. In lines 40-45, as cited in this manuscript, the inherited tectonics has been studied, but, of course, questions remains unsolved in many aspects. Being short of investigation is not a good scientific question for motivation. Instead, I want to see a specific problem in this area, and how this problem can be solved to improve our understanding of the inherited structures, such as how many growth patterns has been studied, how many reactivation modes exist, and etc.

   Done. See Line 46.

4. I am curious on the three stages of extension. Their time spans seem continuous. Is there any evidence on separating them? Fig. 2 provides a stratigraphic column of the study area showing that the earliest stage is compressional and lasted until Early Eocene. It contrasts with the background that there should be an extension stage before Cenozoic. Does this basement have some age constraint? What is the timing of the thrusting?

There are lots of literatures about the tectonic evolution of this area, and the three stages of extension are based on the regional geology. The widely-accepted division scheme of the rifting stage is based on the regional unconformities and sequential rotation of the extension direction recognized in seismic data and drilling data, which correspond to three tectonic events, and separate the syn-rift stratigraphy into three formations. In our study area, however, the present drilling data find that the syn-rift package above the top basement comprises two of them, with the bottom formation undiscoverd, possibly because of the diachronous rifting. Now, we modify the stratigraphic column in Fig. 2 to show that there is a gap between the pre-rift and syn-rift stages.

5. Data acquision is missing in the text. Some introduction can be briefly added in the text.

   Done. See Line 116.

6. I have some questions on the interpretation of seismic profiles. The fault BF2 is partly normal fault and reverse fault. It is clear to identify its upper part, but why is the lower part explained as a reverse fault? I cannot see any evidence of thrusting. Some of red faults in Fig. 5a may be normal fault, such as the first two faults from the right. The authors need more caution on the geometry of faults.

   The reason why we interpret the lower part of BF2 as a reverse fault is that it has similar orientation and dip with the neighbouring basement thrust faults, which are quite common in the study area according to Fig. 5. Yes, the first two faults from the right may be normal faults, but I think it is uncertain because the offset of Tg horizon is too small to be determined with the data resolution. Our point is that they have similar geometry and reflection feature to those neighbouring basement thrust faults, it is more reasonable to make it consistent through the area.

Figures:

7. In fig. 1, the upper sedimentary cover is post-rifting formation, which, however, is cut by some normal faults. There should be a clear definition on the syn- and post rifting stages.

   This is a good point, because post-rift minor fault activity has been reported by previous studies in this and other areas. The boundary between the syn- and post-rifting stages is suggested to be T70 horizon, which has been reached a consensus, but people also

observe that there are some minor tectonic activities during the post-rift stage, causing reactivation of some syn-rift faults (Ye et al., 2017, Earth Sci.).

8. I find some figures are overlapping, Fig. 5a and 6b, Fig. 5b and 9b, and Fig. 5c and 6a. The purpose of emphasizing one segment from a long profile is not clear. Can the short profiles be integrated with the main profiles?

   Yes, the positions of those figures are overlapping, but each of them aims at showing different structural styles with the combination of a few sub-figures. So, integrating them with the main profiles needs to reorganize the order of most of the figures. We think it is OK to just modify them to minimize the overlay, as seen in the figures in the revised manuscript. See figs 5 and 6.

9. The last figure shows different generations of faults. I think some reactivated faults are better marked in red for clarity.

   Done.

Review 2

Below are some major and minor comments:

- Geological framework: in section 2 (lines 88-100) the Authors describe three rifting events but then only two are reported in figure 2, with the collisional phase that overlaps with the older rifting event described in the text. Therefore, the Episode 1 reported in the figure (Early Eocene) does not correspond to the Episode 1 reported in the text (Late Cretaceous-Early Eocene). I suggest the Authors to amend rift names as this may create some confusion.

  I agree with the reviewers that the rifting episodes in the description is confusing, which have also been mentioned by the first reviewer. I will change the name in the revised manuscript. See Section 2.

- Results description: section 4 provides detailed description of observed faults but some parts are slightly mixed with interpretations (e.g., lines 171-184). I would try to keep separated description and interpretations as much as possible and I therefore invite the Authors to consider if these latter may be moved to a separated section.

  I understand your point of separating description from interpretation, so I put the interpretation of the basement structures in a separated section, lines 171-184, after the description of their geometries. I think it is OK to put it in the present place because it is the only part about the intrabasement structure in the manuscript.

- High-angle and low-angle faults definition: the Authors state that "rift-related faults can be divided into low-angle (<30° dip) and high-angle (>30° dip)" but seismic sections are shown in TWT. I therefore wonder what do angles mean, as sections do not appear depth-migrated and fault may therefore bear a different inclination in reality. I therefore invite the Authors to clarify this aspect and to detail how fault angles where measured in the Method section of the manuscript.

  The angle of fault means the dip. We do not measure the exact dip of the faults in seismic sections as they are shown in TWT. However, we are able to know the relative dip of those faults. For example, the cover faults or minor faults developing in the post-rift strata should be dipping at ~60°, so we stretch and compress the sections to let them be approximately dipping at 60° in cross-section view, which appears similar to the sections of depth-migrated. In that way, it is reasonable to determine whether other faults are high-angle or low-angle. See Line 145.

- The Authors have the possibility to measure fault throws and lengths, and in fact they produce T-x plot for a target fault (F1) and T-z profiles for faults F4, F5, F8. The following is not a strict request (Author's interpretations are already well supported), but I wonder if log-log D-L plots (see e.g. Rotevatn, A., Jackson, C.A.L., Tvedt, A.B., Bell, R.E., Blækkan, I., 2019. How do normal faults grow? J. Struct. Geol. 125, 174–184. https://doi.org/10.1016/j.jsg.2018.08.005 or Walsh, J.J., Nicol, A., Childs, C., 2002. An alternative model for the growth of faults. J. Struct. Geol. 24 (11), 1669–1675. https://doi.org/10.1016/S0191-8141(01)00165-1.) may provide further insights on how faults grow end eventually reactivate in the study area. This may be interesting considering that the Authors at line 700-704 talk about "constant length" fault grow model.

  Very good point. We are happy to show more D-L plots of faults in a wider area to the northeast of the present study area for the next paper we are preparing, where there are a lot of reactivated basement faults. That may be a good supplement for this point.

- Paper length: the paper is written in very good English and therefore it is easy to read but it is also quite long and I feel that some parts may benefit from cuts. Particularly, some discussions are quite long and may be shortened. E.g, paragraphs 6.1 and subparagraphs may be largely shortened and even merged into one single section to better focusing on key concepts, the length of some description, despite well written, makes hard to follow the reasoning.

  Yes, we shorten the length of the discussion part in the revision, as also mentioned by the first reviewer.

- Literature: literature is not an issue in this manuscript, as the Authors largely support their findings and discussion with proper references. Nonetheless, some recently published papers may be added to the introduction and further discussed in the manuscript as they may be particularly useful for discussion in section 6.2 and 6.3. I have provided several references in the annotated pdf of the manuscript.

  OK.

Other minor comments and suggestions in the pdf annotated manuscript.

- The introduction part is suggested to cite some recent references.

  Done.

- Line 122. Other wells.

  See Line 125 in the revised manuscript.

- Line 239. Other wells. How do you explain the left-dipping fault depicted in the middle of Fig. 8a crosscutting (or being crosscut??) several fault but showing no sign of displacement?

  We delete the left-dipping fault in F8a.

- Line 258. It is a bit odd to describe isolated faults as a case of fault interaction stating that they do not show interactions...I would rethink at this definition...

  Strictly speaking, isolated fault is not a case of fault interaction, but the goal of presenting it here is to make a comparison with the fault interaction styles, i.e., abutting and merging. It is better for letting readers to know what no interaction looks like, before showing different styles of fault interactions.

- Line 342. unclear as it is written, what do you mean with "shallow dip"?

  Change shallow to low.

- Line 493. I would rather say because you does not have proofs that can exclude it. The fact that this is not the focus os the manuscript (and I agree with this) is not an explanation.

  Done.

- Line 532. see above mentioned reference.

  Done.

- Line 560. I believe that an integration and comparison with updated litearture may reinforce this part.

  The references are added.

---

## Author Response (AR2)

The reviewer recommends the Authors to amend the methodological error and minimize/remove fault angle description/calculation.

As the reviewer recommended, we clarify the shortcoming of the method that we use to estimated the dip of faults, and claim that it does not affect our interpretation of the basement thrusting structures and our focus is the spatial relationship of basement and cover faults, not the actual dips. See Lines 140-143.

We also modify the description of low-angle and high-angle faults in Lines 251-253 of the revised manuscript.

The reference list is modified.